# No compromise in efficiency from the co-application of a marine and a terrestrial CDR method

Yiannis Moustakis [1] ✉, Hao-Wei Wey[2], Tobias Nützel[1], Andreas Oschlies [2,3] & Julia Pongratz[1,4]

Modelled pathways consistent with the Paris Agreement goals to mitigate warming typically include the large-scale application of Carbon Dioxide Removal (CDR), which can include both land- and marine-based CDR methods. However, the Earth system responses and feedbacks to scaling up and/or combining different CDR methods remain understudied. Here, these are assessed by employing two Earth System Models, with a multifactorial setup of 42 emission-driven simulations covering the whole spectrum of Afforestation/Reforestation (0-927 Mha) and of Ocean Alkalinity Enhancement (0-18 Pmol) over the 21$^{st}$ century. We show that global carbon flux responses scale linearly when different CDR methods are scaled up and/or combined, which suggests that the efficiency of CDR is insensitive to both the amount of CDR and the CDR portfolio composition. Therefore, combining CDR methods, which seems beneficial for diversifying risks and remaining below sustainability thresholds, does not compromise the efficiency of individual applications.

Along with stringent emission reductions, Carbon Dioxide Removal (CDR) deployment is required to limit warming to 2 °C, or 1.5 °C relative to pre-industrial levels[1]. A preventive CDR capacity of several hundred $GtCO_2$ is necessary, which would allow for scaling up deployment if needed, to protect and hedge against unexpected high-warming outcomes this century[2]. Given the footprint of CDR on energy-water-land systems, diverse portfolios including both land- and marine-based methods such as Afforestation/Reforestation (AR) and Ocean Alkalinity Enhancement (OAE) should be considered[3], and it is thus crucial to understand how scaling up and/or combining them works within the interactive Earth system.

Besides reducing atmospheric $CO_2$ and thereby global temperatures (biogeochemical cooling), CDR also triggers complex carbon-climate feedbacks[4–7]. In particular, the gradients of $CO_2$ in the air-sea and air-leaf continua are reduced, reducing air-sea $CO_2$ exchange, and photosynthetic uptake respectively[8]. Therefore, enhancing the terrestrial carbon sink reduces the ocean sink, and vice versa, compared to a counterfactual no-CDR scenario. If we thus define the removal efficiency (%) of CDR as the decrease in atmospheric carbon divided by the increase in the land (ocean) carbon following the application of a land- (marine-) based CDR method[9], this should be less than 100%. Such "compensating fluxes" disconnect the carbon sequestration one could theoretically measure in the field from the realized atmospheric $CO_2$ reduction, which has crucial implications for monitoring, reporting, and verifying CDR[8].

Earth System Models of Intermediate Complexity (EMICs) have shown that removal efficiency is strongly dependent on the state of the Earth system and the emissions scenario, whereas the amount of CDR application (and thereby scaling up CDR) is less important[9]. This expectation is based on scenarios where negative emissions are prescribed as permanent removal directly from the atmosphere, and where the removed carbon does not interact with the Earth system[4,10,11]. This setup is representative of methods like Direct Air Carbon Capture and Storage where removed carbon is stored in geological formations and does not interact with the rest of the carbon cycle within the timescales of interest. In AR, carbon sequestration is climate- and $CO_2$-dependent and cannot be a priori known, while the removed carbon remains in the interactive carbon cycle[7]. Planting forests also triggers complex biogeophysical effects by changing the

[1]Ludwig-Maximilians-Universität in Munich, Munich, Germany. [2]GEOMAR Helmholtz Centre for Ocean Research Kiel, Kiel, Germany. [3]Kiel University, Kiel, Germany. [4]Max Planck Institute for Meteorology, Hamburg, Germany. ✉e-mail: yiannis.moustakis@geographie.uni-muenchen.de

properties of the land surface such as albedo and roughness, and thus altering the surface energy and water fluxes[12,13]. Similar considerations apply to OAE, which includes the addition of alkaline materials to the ocean surface, thereby increasing the dissolution of atmospheric $CO_2$ into the sea[14]. The amount of ocean carbon uptake is dependent on factors such as ocean circulation patterns, wind speed, surface ocean carbonate chemistry, and atmospheric $CO_2$ concentrations, which affect gas-exchange kinetics, and cannot be a priori known[15,16].

High-complexity Earth System Models (ESMs), which capture these feedbacks[6,7], are therefore needed to confirm whether the removal efficiency remains insensitive to scaling up and/or combining CDR regardless of the methods employed. Even if this held true, the $CO_2$ removal from combining methods may not be simply the sum of the individual removals, as their combined effect could differ from the linear assumption of independent application. Using an EMIC, Keller et al.[5] suggested that combining AR, OAE, and Ocean Iron Fertilization yields a strongly saturating response of the removal, however this has not been put to the test with an ESM yet. Similarly, scaling up the CDR application by e.g., a factor of two, would not necessarily translate to a doubling of the $CO_2$ removal. Such a linearity would require that carbon uptake per unit of CDR application, is also insensitive to the amount of CDR deployment. Schwinger et al.[15] recently demonstrated that this basically holds true for OAE, but this has not been investigated for AR, or for combined deployments of different CDR methods.

To enhance confidence into our results, we study the scaling up and combination of AR and OAE using two ESMs, namely the Max Planck Institute for Meteorology Earth System Model (MPI-ESM)[17], and the Flexible Ocean and Climate Infrastructure (FOCI)[18], run in an emission-driven configuration where atmospheric $CO_2$ is not prescribed, but dynamically calculated[19]. In particular, we use an AR scenario featuring up to 927 Mha of AR by 2099 following Moustakis et al.[7], and an OAE scenario featuring up to 18 Pmol of OAE over the coastlines globally by 2099, which are also combined and/or halved, while following the fossil fuel emissions of the Shared Socioeconomic Pathway SSP3-7.0 (i.e., a scenario with positive emissions throughout this century). This yields a multifactorial setup of 7 scenarios and 42 simulations in total (Table 1, Fig. 1). We show that global carbon flux responses scale linearly when different CDR methods are scaled up and/or combined, which suggests an insensitivity of the removal efficiency to both the magnitude of the CDR perturbation itself and the CDR portfolio composition.

## Results and discussion
### Scaling up CDR leads to linear responses in carbon fluxes
In the halfAR scenario, land carbon ($C_{land}$) increases by 258 $GtCO_2$ (Figs. 2, S1-2) and $C_{land}$ uptake reaches 56 $GtCO_2$/100 Mha by 2099

(Figs. 3, S3–4). An increase of $C_{land}$ uptake/100 Mha across time could be expected given the cumulative effect of continuous carbon sequestration by forest planted early on, which is dependent on the characteristics of re/afforested land. Here, reforestation is prioritized over afforestation, and thus land with higher sequestration potential is converted early on[7]. However, this increase mostly reflects the effect of increasing atmospheric $CO_2$ concentrations on photosynthesis[20]. The dependency on $CO_2$ concentrations becomes evident when comparing with the findings of Moustakis et al.[7], who used the same setup albeit under the lower SSP5-3.4os emissions, and reported a lower value of ~40 $GtCO_2$/100 Mha by 2100.

Doubling the amount of AR deployment results in a minor decrease in $C_{land}$ uptake/100 Mha reaching 54 $GtCO_2$/100 Mha (Figs. 3, and S3–4 for model-specific results), and thus $C_{land}$ scales linearly reaching 503 $GtCO_2$ (Fig. 2), which is only ~3% lower than linear expectations (2*halfAR), with linearity holding across time (Figs. S5–7). This could be expected, given that all land use transitions are halved, and thus the halfAR and AR scenarios contain identical fractions of converted productive and less productive land[7]. However, in the AR scenario atmospheric $CO_2$ levels are ~27 ppm less by 2099 compared to halfAR, which could suggest a lower $CO_2$ effect on forests. Still, this difference only gradually builds up, whereas ambient $CO_2$ levels in all scenarios by the end of the century are in a range (>700 ppm) where the slope of photosynthetic gain per ppm of $CO_2$ increase in JSBACH3 (the land surface model of both ESMs employed here, see Methods) is rather saturated[21]. Therefore, the additional loss of fertilization in the AR scenario as atmospheric $CO_2$ levels decrease is weak, and seems to be compensated for by the resulting carbon-climate feedbacks and the associated stronger biogeochemically-induced cooling[22].

In the halfOAE scenario, ocean carbon ($C_{ocean}$) increases by 262 $GtCO_2$ (Figs. 2, S1–2), and the $C_{ocean}$ uptake/Pmol by ~0.14 $GtCO_2$/Pmol for every 10 ppm increase in atmospheric $CO_2$, reaching 33 $GtCO_2$/Pmol by 2099 (Figs. 3, S3–4). Given this sensitivity and the ~30 ppm difference between halfOAE and OAE, scaling up OAE does not substantially affect $C_{ocean}$ uptake/Pmol. Lenton et al.[23] applied the same amount of alkalinity under the Representative Concentration Pathway RCP8.5 and RCP 2.6 and reported ~34 and ~26 $GtCO_2$/Pmol respectively, also suggesting that $C_{ocean}$ uptake/Pmol increases with increasing atmospheric $CO_2$ concentrations[15]. Despite this, using MPI-ESM (CMIP5 version) González & Ilyina[24] reported a lower uptake (~30 $GtCO_2$/Pmol) than the one reported here, even though emissions were higher (RCP8.5). However, this followed the application of 114 Pmol of alkalinity, which is significantly higher than the 8 and 16 Pmol applied here. In fact, Feng et al.[25] reported that over an extremely wide range of OAE applications spanning from 5 to 151 Pmol, uptake decreases from 32 to 27 $GtCO_2$/Pmol.

**Table 1 | The table shows the detailed characteristics of the employed Carbon Dioxide Removal (CDR) scenarios featuring Afforestation/Reforestation (AR) and Ocean Alkalinity Enhancement (OAE) and how linear expectations are formulated**

| Scenario | AR by 2099 (Mha) | Alkalinity by 2099 (Pmol) | Assess linearity compared to: | Removal Efficiency (%) | $\frac{C_{land}\,uptake\,(GtCO_2)}{100\,Mha}$ | $\frac{C_{ocean}\,uptake\,(GtCO_2)}{Pmol}$ |
|---|---|---|---|---|---|---|
| **REF** | – | – | – | – | – | – |
| **halfAR** | 463.5 | – | – | $\frac{\Delta C_{atmo}^{halfAR-REF}}{\Delta C_{land}^{halfAR-REF}}$ | $\frac{\Delta C_{land}^{halfAR-REF}}{Mha/100}$ | – |
| **AR** | 927 | – | 2 * halfAR | $\frac{\Delta C_{atmo}^{AR-REF}}{\Delta C_{land}^{AR-REF}}$ | $\frac{\Delta C_{land}^{AR-REF}}{Mha/100}$ | – |
| **halfOAE** | – | 8 | – | $\frac{\Delta C_{atmo}^{halfOAE-REF}}{\Delta C_{ocean}^{halfOAE-REF}}$ | – | $\frac{\Delta C_{ocean}^{halfOAE-REF}}{Pmol}$ |
| **OAE** | – | 16 | 2 * halfOAE | $\frac{\Delta C_{atmo}^{OAE-REF}}{\Delta C_{ocean}^{OAE-REF}}$ | – | $\frac{\Delta C_{ocean}^{OAE-REF}}{Pmol}$ |
| **halfMixed** | 463.5 | 8 | halfAR + halfOAE | – | $\frac{\Delta C_{land}^{halfMixed-REF}}{Mha/100}$ | $\frac{\Delta C_{ocean}^{halfMixed-REF}}{Pmol}$ |
| **Mixed** | 927 | 16 | (1) 2 * halfMixed (2) AR + OAE | – | $\frac{\Delta C_{land}^{Mixed-REF}}{Mha/100}$ | $\frac{\Delta C_{ocean}^{Mixed-REF}}{Pmol}$ |

The table also shows how the removal efficiency and carbon uptake per unit of CDR application are expressed as a function of changes in atmospheric ($C_{atmo}$), land ($C_{land}$), and ocean ($C_{ocean}$) carbon. For every scenario, 3 ensemble members for each Earth System Model (ESM) are available, spanning from 2015 to 2099.

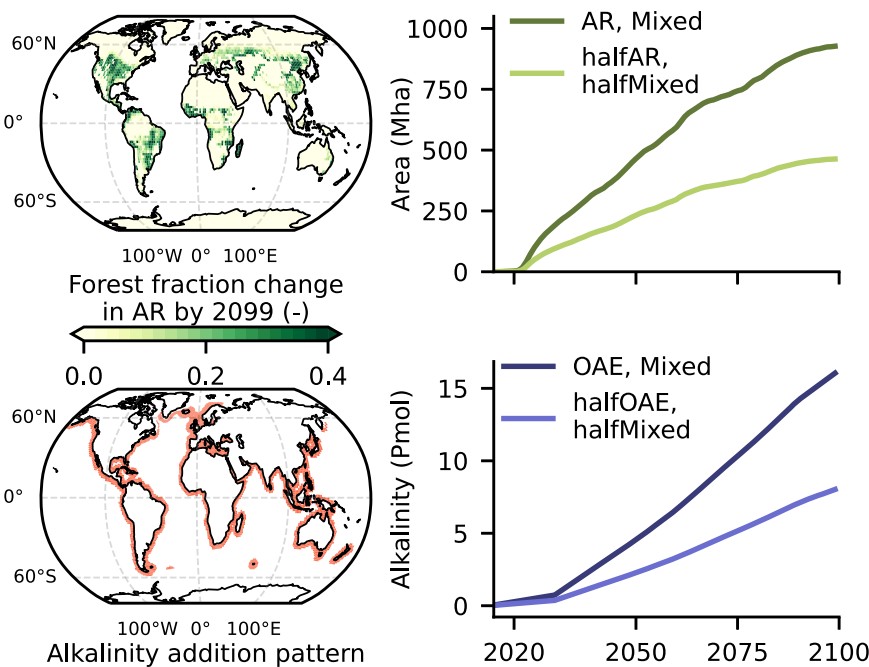

**Fig. 1 | Simulation forcings.** (Top) The map shows the Afforestation/Reforestation (AR) pattern expressed in forest cover fraction change by 2099 in the AR/Mixed scenarios. For the halfAR/halfMixed scenarios, the spatial pattern is the same, albeit halved. The line plot shows the global forest area increase (in Mha) across time. The details of the pattern can be found in Moustakis et al.[7] (Bottom) The map shows the coastline gridcells where alkalinity is added. The line plots show the cumulative amount of alkalinity addition (in Pmol) across time.

Given the above, unless vastly different application rates are considered, $C_{ocean}$ uptake/Pmol can be considered insensitive to scaling up OAE, and dependent on ambient $CO_2$ concentrations. This leads to a linear $C_{ocean}$ increase[26] holding across time (Figs. S5–7), and reaching 521 GtCO2 in the OAE scenario (Figs. 2, S1–2), which differs by <1% from linear expectations (2*halfOAE).

Overall, our results suggest that the carbon uptake per unit of CDR application is rather insensitive to scaling up the application, and therefore the land (ocean) carbon flux responses in AR (OAE) are linear, since even ambitious deployment -as is the case here- impacts atmospheric $CO_2$ by amounts small compared to those of different emission scenarios. Future studies should investigate to what extent linearity holds for scenarios featuring strongly reduced emissions. Under lower atmospheric $CO_2$ concentrations, the $C_{land}$ and $C_{ocean}$ uptake might show less muted responses to CDR-induced changes in atmospheric $CO_2$, which might distort the linearity.

### Removal efficiency is insensitive to scaling up CDR
In halfAR and AR the decrease in atmospheric carbon ($C_{atmo}$) reaches 220 and 429 GtCO2 respectively by 2099 (Figs. 2, S1–2), yielding a removal efficiency of 85% in both scenarios and both models (Figs. 3, S3–4), which is the result of $C_{ocean}$ compensating fluxes mostly over the Southern Ocean (Figs. 4, S8–9). This is significantly higher than the 74% reported in Moustakis et al.[7] under SSP5-3.4os emissions, suggesting that removal efficiency increases with increasing emissions[4]. Using MPI-ESM (CMIP5 version), Sonntag et al.[27,28] reported a 83% removal efficiency under RCP8.5 following a $C_{land}$ increase of 793 GtCO2 (Table S1). Loughran et al.[29] reported 95% under SSP5-8.5 with ACCESS-ESM1-5 after a $C_{land}$ increase of 92 GtCO2 (Table S1). Using HadGEM2-ES, Koch et al.[30] reported a significantly lower removal efficiency under RCP2.6 reaching 55%, following a $C_{land}$ increase of 121 GtCO2 (Table S1). Wey et al.[6] reported a range of 75–88% across 7 ESMs under SSP5-8.5 (2040-2060 average), following a $C_{land}$ increase of 113 ± 30 GtCO2 (multi-model mean ± one standard deviation) (Table S1).

In the halfOAE and OAE scenarios, the decrease in $C_{atmo}$ reaches 225 and 455 GtCO2 respectively by 2099 (Figs. 2, S1–2), translating to a removal efficiency of 86 and 87% (Figs. 3, S3–4). This is the result of the $C_{land}$ compensating fluxes over eastern Asia, the U.S.A., and the tropics (Figs. 4, S8–9). Recently, Jeltsch-Thömmes et al.[19] reported a removal efficiency of 73% with UVic EMIC under SSP5-3.4os and 87% with the Bern3D v2.0 EMIC by 2100 (Table S1). The emerging insensitivity contradits Palmiéri and Yool[26], who showed with UKESM that under SSP5-8.5 emissions halving or doubling OAE application rates, or changing the depth of application, causes the removal efficiency to strongly vary from 57 to 103%, despite the linear $C_{ocean}$ increase (Table S1). This likely is due the lack of ensemble members therein, because in all cases but one, the CDR perturbation is small (<100 GtCO2), and internal variability likely masks the signal. This agrees with our findings showing that removal efficiency is highly variable by ~2050, when $C_{atmo}$ reduction is still weak (Fig. 3).

Wey et al.[6] reported a range of 84–91% across 4 ESMs under SSP5-8.5 (2040-2060 average), following a $C_{ocean}$ increase of 128 ± 2 GtCO2 (multi-model mean ± one standard deviation) (Table S1). Sonntag et al.[28] reported a removal efficiency of 96% under RCP8.5 with MPI-ESM (CMIP5 version) following a $C_{ocean}$ increase of ~3453 GtCO2 by 2099 (Table S1). Under RCP8.5, Lenton et al.[23] reported an average removal efficiency of 98%, which is reduced to 84% under RCP2.6 (Table S1). Therefore, similar to the case of AR, removal efficiency likely depends on the emission scenario when OAE is applied. Feng et al.[25] reported a removal efficiency between 85% and 92% with UVic EMIC, and showed that it decreased linearly with increasing amounts of $C_{ocean}$ uptake (Table S1). However, this included up to 151 Pmol of alkalinity, and our results suggest that for a smaller range of alkalinity addition (up to 16 Pmol), the removal efficiency is rather insensitive to increasing application rates. Nevertheless, uncertainty remains regarding the magnitude of CDR perturbation that can be considered strong enough to affect the removal efficiency. For example, Jones et al.[4] reported small variations up to 5% when quadrupling the magnitude of CDR, despite it being even bigger than the cumulative emissions in the underlying RCP2.6 scenario.

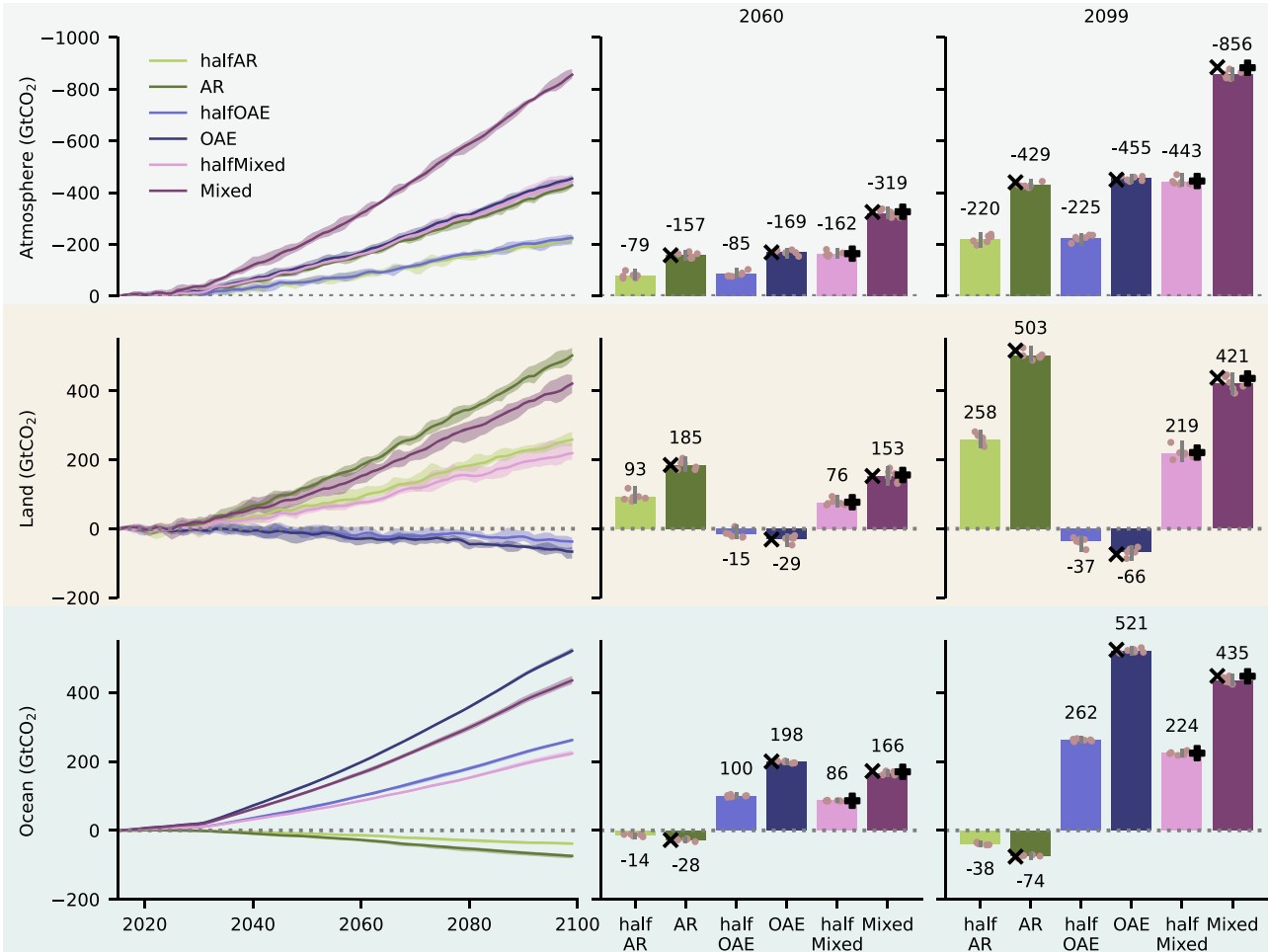

**Fig. 2 | Carbon sequestration.** The left column panels show the timeseries of intermodel average change in (top) atmosphere, (middle) land, and (bottom) ocean carbon for all the different Carbon Dioxide Removal (CDR) scenarios compared to REF (GtCO₂). The shading around the mean shows the minimum-maximum range across both models and all ensemble members. To aid interpretation, in the middle and right column panels the barplots show snapshots for 2060 and 2099. The bar height corresponds to the intermodel average, and the gray vertical lines show the minimum-maximum range across both models and all ensemble members, while the individual data points are also plotted. The multiplication and plus signs indicate expectations from scaling up CDR application and combining methods respectively, based on linearity assumptions (see Table 1). To aid visualization, the vertical axis is flipped in the top panel.

Given the above, our results suggest that the removal efficiency is rather insensitive to the rate of application for both methods. Therefore, the linearity of carbon flux responses across time is due to the emerging insensitivity of both the carbon uptake per unit of CDR application and the removal efficiency to scaling up CDR.

### Is OAE more efficient than AR in removing carbon?

Our setup allows for a direct comparison of the removal efficiency between the two methods, which in OAE is slightly higher by 1–2% (Figs. 3, S3–4). When OAE is applied, the land surface has overall less forest cover compared to the AR scenario, and thus the land has less capacity to sequester carbon and compensate for the increasing $C_{ocean}$. Therefore, in the AR and OAE scenarios the Earth system is in different states that may offer different capacities for redistributing carbon and compensating. This contradicts Sonntag et al.[28], who reported a 96% removal efficiency for OAE and 83% for AR. However, their study included vastly different amounts of CDR, featuring a 793 GtCO₂ $C_{land}$ increase for AR, and a 3453 GtCO₂ $C_{ocean}$ increase for OAE (Table S1). Similarly, Keller et al.[5] reported a removal efficiency of 80% for AR (480 GtCO₂ $C_{land}$ increase) and 92% for OAE (664 GtCO₂ $C_{ocean}$ increase) using UVic EMIC under RCP8.5, despite the $C_{land}$ and $C_{ocean}$ uptake amounts not being vastly different.

It should be noted that the removal efficiency metric is more indicative of the strength of the land (ocean) compensation as a response to a given increase in $C_{ocean}$ ($C_{land}$) rather than the efficiency of the CDR method itself. This is due to the fact that the net change in a sink where a CDR method is applied includes not only the carbon uptake through CDR itself, but also other concurrent compensating fluxes over that given sink. For example, in the halfAR and AR scenarios, a consistent weakening of the Amazon carbon sink is obtained in both models (Figs. 4, S8–9). Similarly, even though ocean carbon uptake over the coastlines emerges clearly when OAE is applied, concurrent compensating fluxes over the Southern Ocean are evident in both models, especially in the OAE scenario (Figs. 4, S8–9).

Strictly separating the CDR sequestration from the total change in a sink is not a trivial task, and would require additional simulations or model development. For OAE, in both models added alkalinity is naturally transported to adjacent gridcells resulting in increased CO₂ uptake, for example across the Bay of Bengal and Western Indian Ocean, and the South Atlantic Ocean (Figs. 4, S8–9). With our simulations this cannot be separated from CO₂ uptake over the gridcells of OAE application. Additionally, the carbon increase in an ocean gridcell where OAE is applied also depends on changing climate and atmospheric CO₂ levels[31], and isolating the naturally occurring flux would require additional concentration-driven simulations[15]. Similarly for AR,

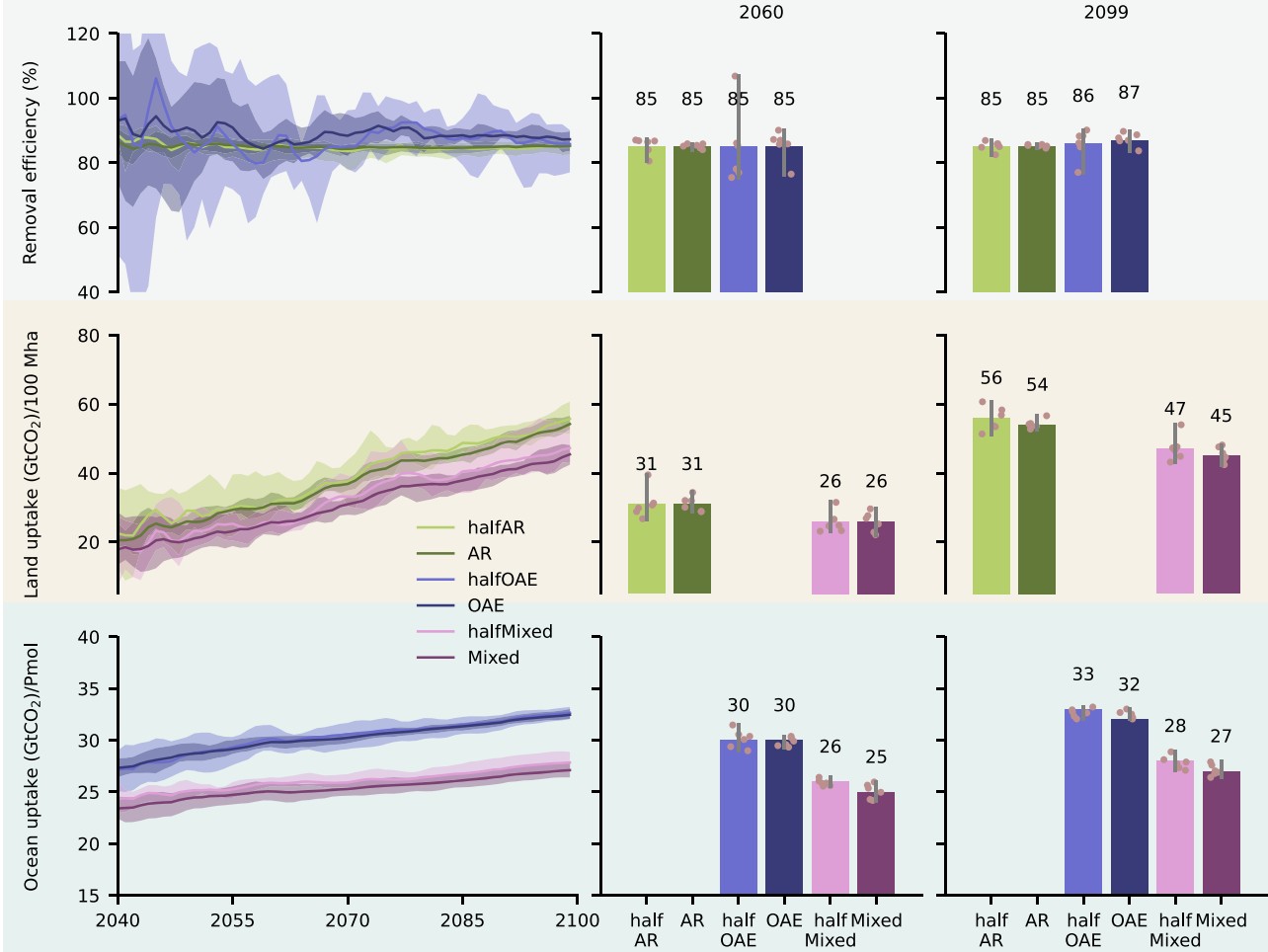

**Fig. 3 | Removal efficiency and uptake per unit of Carbon Dioxide Removal (CDR) application.** The left column panels show the timeseries of intermodel average: (top) removal efficiency (see Methods, Table 1), (middle) land carbon uptake ($GtCO_2$) per 100 Mha of planted forest, and (bottom) ocean carbon uptake ($GtCO_2$) per Pmol of alkalinity added to the ocean surface. In all cases, uptake and removal refer to changes in carbon stocks compared to REF. The shading around the mean shows the minimum-maximum range across both models and all ensemble members. To aid interpretation, in the middle and right column panels the barplots show snapshots for 2060 and 2099. The bar height corresponds to the intermodel average, and the gray vertical lines show the minimum-maximum range across both models and all ensemble members, while the individual data points are also plotted.

the carbon fluxes of the new forest cannot be separated from already standing forest, which naturally responds to AR, as also do all vegetation types in a gridcell where AR is applied. Separating the carbon sequestered through CDR would thus require a configuration separating the new forest, and concentration-driven runs aimed at quantifying the feedback from reduced atmospheric $CO_2$ concentrations.

Given the above, we argue that despite removal efficiency in OAE being slightly higher, this does not suggest that OAE reduces $C_{atmo}$ more efficiently than AR.

**Linear carbon-cycle responses to combining CDR methods**

Our results suggest that the global carbon fluxes scale linearly also when combining CDR methods (Figs. 2, S5–7). In the Mixed scenario, $C_{atmo}$ reduction reaches 856 $GtCO_2$ by 2099 (Figs. 2, S1–2), due to the $C_{land}$ and $C_{ocean}$ increase of 421 and 435 $GtCO_2$ respectively. In the halfMixed scenario $C_{atmo}$ reduction is 443 $GtCO_2$, due to the $C_{land}$ and $C_{ocean}$ increase of 219 and 224 $GtCO_2$ respectively (Figs. 2, S1–2). $C_{atmo}$ reduction in the halfMixed and Mixed scenarios is ~0.4% and ~3% less than linear expectations, respectively. This is rather negligible and is dominated by the variability of the terrestrial carbon sink (Figs. S5–7). However, it could point towards a tendency of the Earth system to yield potentially saturating responses in case strong enough perturbations are imposed to the Earth system, thus starkly reducing

ambient atmospheric $CO_2$ levels, as discussed above. Our results suggest that at the global level there is little interaction between AR and OAE, despite the biogeophysical effects of AR on hydroclimatic variability and potential changes in the freshwater flux into the ocean, which are rather lower-order effects. The linearity of responses reported here contradicts Keller et al.[5], which is the only study that has investigated combining CDR methods so far, albeit with an EMIC. In particular, they reported that combining AR, OAE, and Ocean Iron Fertilization under RCP8.5 yields by 2100 a $C_{atmo}$ reduction which is ~23% (308 $GtCO_2$) less than linear expectations. This saturation does not hold for 2030, when $C_{atmo}$ reduction is only ~5% (22 $GtCO_2$) less. Their overall perturbation to the Earth system includes a ~19% higher $C_{atmo}$ reduction than the one reported here, reaching ~1017 $GtCO_2$ by 2100, but we cannot assess whether this difference can be a reason for disagreement. Notably, in their study, the strongest deviation from linearity is obtained for the $C_{ocean}$, yielding 33% and 35% less by 2030 and 2099 respectively, whereas in our study the ocean emerges as the most strongly linear and less variable sink in both models. This could imply that the inclusion of Ocean Iron Fertilization distorts linearity, but further research is needed.

Importantly, the insensitivity of carbon uptake per unit of CDR application to scaling up application rates holds not only in the (half) AR and (half)OAE scenarios, but also between the halfMixed and Mixed

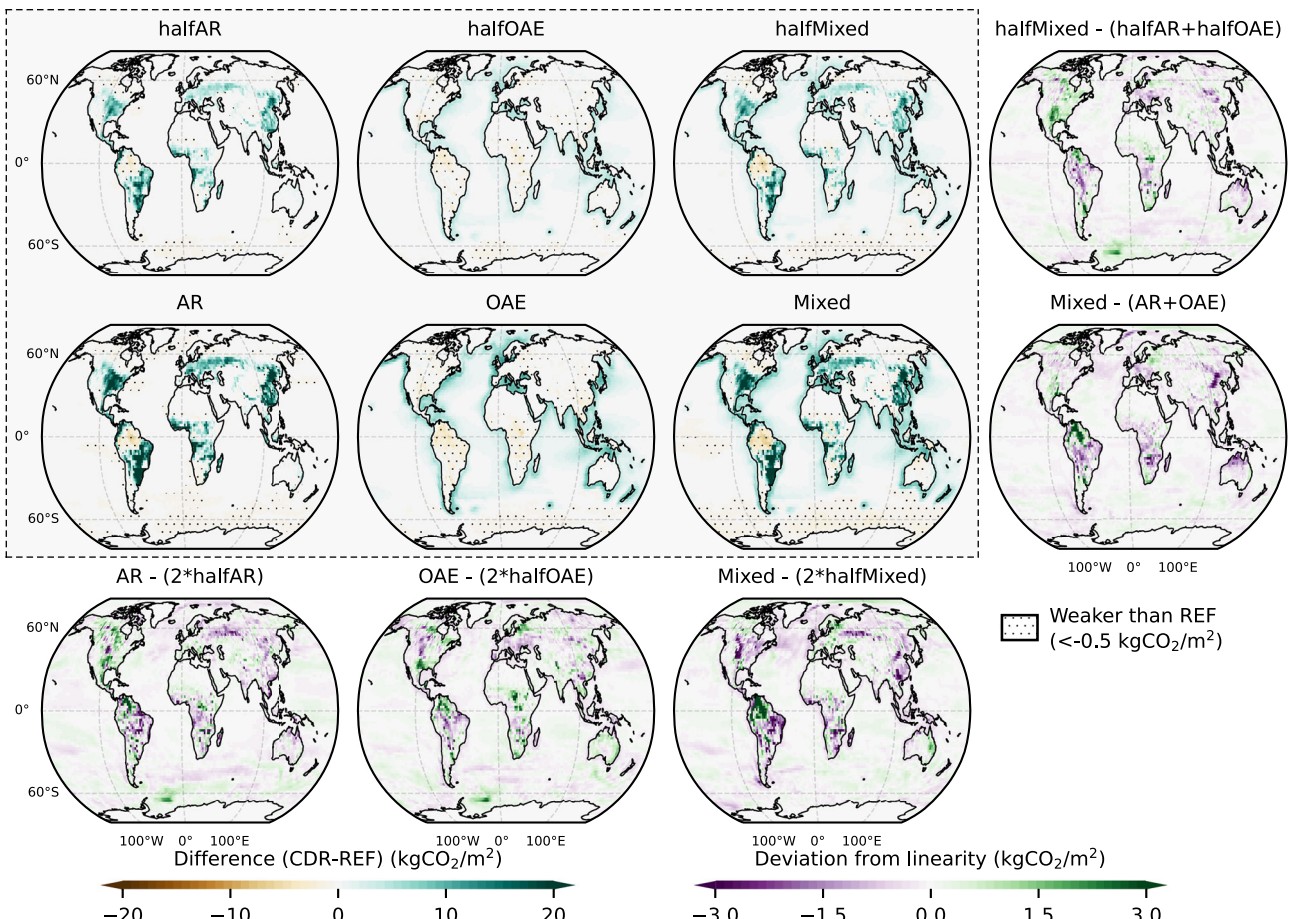

**Fig. 4 | Carbon fluxes at the gridcell level.** The maps within the shaded box show the intermodel average difference in cumulative carbon flux ($kgCO_2/m^2$) between each Carbon Dioxide Removal (CDR) and the REF scenario by 2099. A positive value (shown in blue) indicates carbon sequestration on land and ocean compared to REF, while a negative value (shown in brown) suggests that the land or ocean carbon sink is reduced compared to REF. To aid interpretation, the hatching highlights the regions with a negative value less than $-0.5$ $kgCO_2/m^2$. The maps outside the shaded box show the deviation from linear expectations (see Table 1),
as noted in the titles. A different scale and colormap are used, with positive values (shown in green) indicating that the realized flux (sequestration or weakening compared to REF) is higher than the linear expectation (sequestration or weakening compared to REF) (see Table 1). For example, in gridcells where both the realized and expected flux are negative (weakening compared to REF), a positive value suggests that the realized weakening is less strong in magnitude as the expected one.

scenarios. Therefore, linearity holds for scaling up CDR methods even in the portfolio case (Figs. S5–7). In particular, $C_{land}$ uptake/100 Mha in the halfMixed and Mixed scenarios reaches 47 and 45 $GtCO_2$/100 Mha, respectively, while $C_{ocean}$ uptake/Pmol is 28 and 27 $GtCO_2$/Pmol (Figs. 3, S3–4). Even though the carbon uptake per unit of CDR application for both methods is lower in the (half)Mixed scenarios than their single-CDR counterparts, this is not due to differences in atmospheric $CO_2$ concentrations. This becomes evident in the halfMixed scenario, where the carbon uptake per unit of CDR application for AR and OAE is still lower than the values obtained in the AR and OAE scenarios, respectively, despite the trajectory of atmospheric $CO_2$ being similar across all three scenarios.

Even though the CDR sequestration cannot be isolated, there is no consistent reduction across models in sequestered carbon over sites of forestation or over the ocean gridcells where OAE is applied when the (half)AR and (half)OAE are combined (Figs. 4, S8–9). This implies that it is rather the emerging compensating fluxes that reduce the AR and OAE carbon uptake per unit of CDR application under the (half)Mixed scenarios, and not the capacity of each method itself in sequestering carbon locally, which is insensitive to small changes in ambient $CO_2$ concentrations. For example, the weakening of the Amazon and Southern Ocean sinks is also evident in the (half)Mixed scenarios in both models (Figs. 4, S8–9).

As a result, the emerging linearity of the carbon fluxes under the (half)Mixed scenarios suggests a linear behavior of compensating fluxes when combining individual CDR methods. This would mean that the insensitivity of removal efficiency to the amount of CDR perturbation holds not only when scaling up an individual CDR method, but also when a perturbation is induced by the introduction of a different method. This suggests that the removal efficiency of CDR is not compromised at the portfolio case. Even though the removal efficiency cannot be quantified here, there is no apparent reason indicating that this should not hold. Nevertheless, future studies should further investigate to what extent it is the CDR sequestration itself, the compensating fluxes, or both, that facilitate linearity. Future studies should further validate our results by exploring diverse portfolios that incorporate a broader range of CDR methods, assessing whether interactions between specific methods can occur, potentially offsetting linearity.

**Implications of linearity for project-level estimates**
Here, we argue that despite the emerging linearity of global and regional carbon fluxes, caution is needed when considering project-level estimates. Deviation from linearity is evident locally (Figs. 4, S8–9) and is stronger over land, due to the emerging complex, non-linear local and non-local feedbacks especially when AR is considered[8,12,32,33], but is still

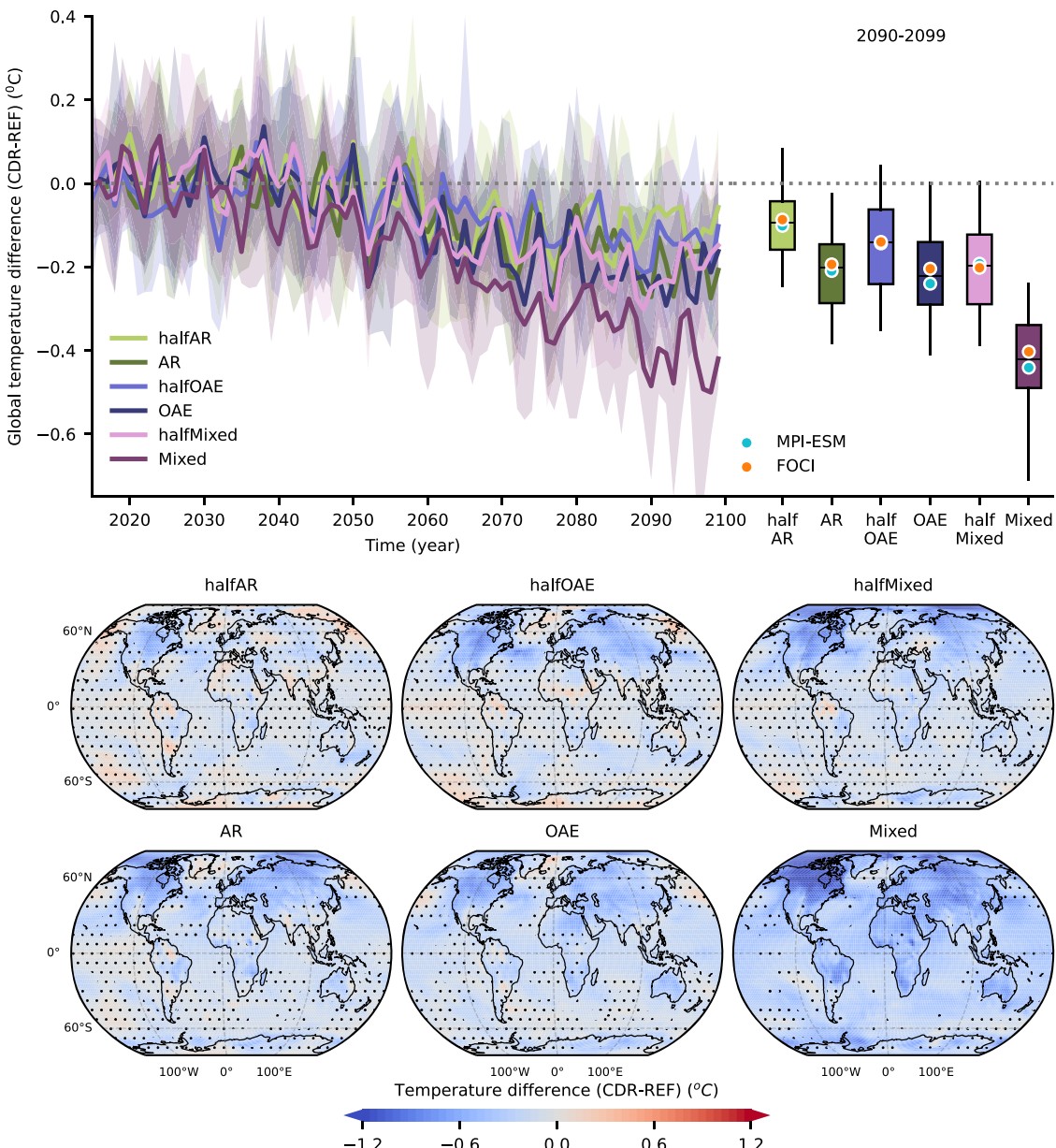

**Fig. 5 | Warming mitigation.** (Top) For every Carbon Dioxide Removal (CDR) scenario the lineplots show the timeseries of the intermodel average difference in globally averaged 2 m temperature (°C) compared to the mean temperature under REF. The shading around the mean shows the minimum-maximum range across both models and all ensemble members. To aid interpretation, the boxplots on the right show yearly values of globally averaged temperature between 2090 and 2099 pooled from both models and all ensemble members. The whiskers show the 5th–95th percentiles of pooled values. The model-specific 2090–2099 averages are shown with the colored circles, and the intermodel average with the horizontal black lines. (Bottom) The maps show the intermodel average difference in average 2090–2099 2 m temperature (°C) between each CDR and the REF scenario. Stippling highlights statistical insignificant differences. Statistical significance is declared over gridcells where: (a) the two models agree on the sign of the change, and (b) at least one model shows a statistically significant difference at the 10% significance level based on a two-tailed Student's $t$ test adjusted to account for temporal lag-1 autocorrelation[67]. For each gridcell and model, the test is applied over the yearly data pooled together from all ensemble members for the given period.

small compared to the amounts of carbon sequestered, and does not have distinct spatially organized features that are consistent across models. For example, the models disagree over the linearity of fluxes across the forestation sites in Asia after scaling up AR, with FOCI showing a more consistent less-than-linear flux (Figs. S8–9). Similarly, scaling up AR results in more than linearly scaled sequestration over the U.S.A. in MPIESM (Figs. S8–9). The uncertainty of $C_{land}$ fluxes becomes evident for both models, by the apparent differences between the two patterns of divergence from linearity that are available for the Mixed scenario. Therefore, sequestration estimates for individual forestation projects obtained by multiplying fixed forest carbon densities with the

area of application cannot yield accurate results. Even though such estimates could still likely serve as a first order approximation, they should be treated with caution. Since the dependency of the $C_{land}$ uptake/100 Mha on the background climate and $CO_2$ trajectory is strong, approaches that use transient forest carbon densities to estimate carbon accumulation based on future climate and $CO_2$ levels should be preferred[34].

$C_{ocean}$ fluxes in the (half)OAE scenarios scale linearly also locally, especially over the regions where the bulk of sequestration has occurred (Figs. 4, S8–9). This is consistent across both models, despite in FOCI a higher ocean resolution being employed, adding confidence

to our results. Given the difficulty in directly monitoring verifying and reporting OAE[35], our modeling results imply that linear estimates could be useful for rough first order approximations of OAE uptake when scaling up application in real life. However, model estimates for OAE can still be uncertain and require further experimental validation[35].

It should be noted that the linearity reported here emerges after increasing the magnitude of CDR application over the exact same gridcells. However, this might not be the case when scaling up the CDR deployment includes applying the additional alkalinity and/or establishing the additional forests over different regions. This is due to the fact that different regions can yield different rates of $C_{ocean}$ uptake/Pmol for OAE[16] and of $C_{land}$ uptake/100 Mha for AR, since forests can have different capacities to sequester carbon across different regions[7,36]. However, when it comes to removal efficiency, our results suggest that it is insensitive to the magnitude of the CDR perturbation and there is no apparent reason why this should not hold in cases where scaling up CDR includes different regions of deployment. Overall, increasing the model's resolution could improve the representation of climatic variability[37], thus offering useful insight into carbon flux responses over land gridcells, where deviation from linearity is stronger (Fig. 4). However, a coarser resolution (as is the case here) can still offer qualitatively similar results with little biogeochemical error compared to higher resolutions[38], thus capturing the (non)linearity of $C_{land}$ responses over land gridcells on average.

## Warming mitigation

Here, similar levels of CDR amounts are applied under the same carbon emissions, which suggests that the sensitivity of global temperature to changes in atmospheric $CO_2$ levels is similar in all scenarios[39]. In the halfAR and AR scenarios, average 2090–2099 temperature is reduced by $0.09 \pm 0.1\,°C$ (mean ± standard deviation, estimated by pooling yearly data) and $0.2 \pm 0.11\,°C$ respectively (Figs. 5, S10–11), which corresponds to ~0.05 °C warming mitigation per 100 GtCO2 of $C_{atmo}$ reduction (considering $C_{atmo}$ by 2095), matching the climate sensitivity of the REF experiment. In both scenarios and models, even though biogeophysically-induced warming likely partly offsets the overall stronger biogeochemical cooling[7], warming does not strongly emerge locally (Figs. 5, S10–11). In halfAR, cooling emerges over North America, Africa, and Latin America, while cooling is widespread in the AR scenario in both models. Efficiency in mitigating warming is lower than the 0.07 °C/100 GtCO2 in Moustakis et al.[7], since the SSP5-3.4os scenario they employed has lower atmospheric $CO_2$ levels, and thus a higher sensitivity of the radiative forcing to changes in atmospheric $CO_2$ concentrations[39]. Previous studies have suggested similar levels of warming mitigation following large-scale AR application. Using MPI-ESM (CMIP5 version), Sonntag et al.[27] reported a 0.27 °C cooling under RCP8.5 following 800 Mha of AR and 100 Mha of avoided deforestation. Under high emissions, Arora & Montenegro[40] showed a 0.25 °C warming mitigation with an ESM, following 1,010 Mha of AR. Dooley et al.[41] reported a warming mitigation up to 0.25 °C under a low overshoot scenario. Following SSP5-8.5 emissions and SSP1-2.6 land use pattern featuring large-scale AR, Loughran et al.[29] reported no effect of temperature based on 6 ESMs for a resulting 37-220 GtCO2 $C_{atmo}$ reduction range.

In the halfOAE and OAE scenarios a widespread cooling signal occurs in both models, and warming mitigation reaches $0.14 \pm 0.13\,°C$ and $0.22 \pm 0.12\,°C$, respectively (Figs. 5, S10–11), corresponding to an efficiency in mitigating warming of 0.07 and 0.05 °C /100 GtCO2. This is higher than the 0.04 °C /100 GtCO2 based on Keller et al.[5], and the 0.02 °C /100 GtCO2 based on Lenton et al.[23], albeit under higher RCP8.5 emissions. Other studies have reported cooling of even ~1.5 °C, but following vastly higher total alkalinity addition to the ocean[24,28]. Even though the absence of AR-induced biogeophysical effects on temperature could imply that OAE is more efficient in mitigating warming, our results do not show a statistically significant difference between AR and OAE, or halfAR and halfOAE in both models, and larger ensemble sizes would be required for any signal to robustly emerge. In the halfMixed and Mixed scenarios, warming mitigation reaches $0.20 \pm 0.13\,°C$ and $0.42 \pm 0.14\,°C$ respectively (Figs. 5, S10–11), corresponding to an efficiency in mitigating warming of 0.05 °C/100 GtCO2. In the halfMixed scenario, widespread cooling occurs over parts of the Northern hemisphere, the tropics, and Antarctica, while in the Mixed scenario, cooling is dominant globally in both models (Figs. 5, S10–11).

Overall, our results suggest that scaling up and/or combining CDR methods yields a roughly linear increase in the mitigation of global warming, which is the result of the nearly constant transient climate response to cumulative net positive $CO_2$ emissions[42]. However, this does not hold at the gridcell-level, where the complex non-linear dynamics of both the biogeophysical and biogeochemical effects of CDR application on surface energy and moisture fluxes are at play[32,33,43]. Nevertheless, examining model-specific results (Figs. S10–11) suggests that patterns of biogeophysically-induced warming tendencies -even though statistically insignificant- tend to emerge also after scaling up AR or combining it with OAE, and are expressed either as a net warming or a weaker cooling (e.g., warming tendency over the Sahel region in MPI-ESM (Fig. S10)). This suggests that, despite temperature responses being non-linear, such features can still persist when scaling up and/or combining CDR methods. Nevertheless, in the absence of large ensemble sizes, temperature responses at the gridcell level can be masked by internal variability.

## Outlook

Our study is a thorough attempt to disentangle the dynamics of CDR portfolios, by employing two ESMs. Global carbon fluxes respond linearly to CDR perturbations, due to the insensitivity of the carbon uptake per unit of CDR application and the removal efficiency to both the magnitude of the CDR perturbation, and the portfolio composition[44]. Our results suggest greater flexibility in designing sustainable CDR portfolios that incorporate both land- and marine-based CDR methods, since combining methods does not compromise the removal efficiency of individual applications globally, even in the presence of emerging feedbacks. This flexibility can be advantageous for managing risks and ensuring that future CDR deployment remains within ecologically and socially acceptable levels[3,45].

With CMIP7 on the horizon, and as IAMs and ESMs are introducing more CDR methods[36,46], we call for an increased focus on diverse CDR portfolios and the emerging carbon-climate feedbacks, which can only be facilitated by including more emission-driven simulations[44]. Apart from validating our results, future studies should also employ setups tailored to isolating CDR sequestration from compensating fluxes, and investigate other trajectories as well, especially overshoot ones, which can feature complex sink-to-source transitions and asymmetries[4,11,47].

## Methods

### Models employed

We employ two coupled ocean-land-atmosphere ESMs, namely MPI-ESM (MPI-ESM-1-2.01p7-LR)[17], and FOCI[18]. MPI-ESM has been participating in the Coupled Model Intercomparison Projects including phase 6 (CMIP6), and has been widely applied, studied, and evaluated against observations and other ESMs[48–50]. MPI-ESM has already been used for studies including AR and OAE[7,24,27,28,51]. MPI-ESM employs ECHAM6[52] as the atmospheric component with a T63 (1.9°) horizontal resolution and 47 vertical atmospheric layers, JSBACH3[53] as the land component with a T63 (1.9°) horizontal resolution, MPIOM[54] as the ocean component employing a bipolar grid with 1.5° resolution and 40 vertical layers, and HAMOCC6[55,56] as the marine biogeochemical component. FOCI is the successor of the Kiel Climate Model[57] and employs ECHAM6 as the atmospheric component with a T63 (1.9°) horizontal resolution and 95 vertical atmospheric layers, JSBACH3 as the land component with a T63 (1.9°) horizontal resolution, NEMO[58] as

the ocean component employing a tripolar ORCA05 grid with a resolution of 0.5°, which corresponds to 55.6 km near the equator and 46 vertical layers, and MOPS[59] as the marine biogeochemical component.

Given the above, FOCI and MPI-ESM feature the same land and atmospheric components, albeit with increased vertical atmospheric resolution in FOCI. The two models have different ocean and marine biogeochemical components, while FOCI has also a higher horizontal and vertical ocean resolution. Higher ocean resolution is crucial for properly capturing alkalinity diffusion, transport and mixing towards deeper sea layers and adjacent areas[60], and this has motivated us to create a mini-ensemble of two models that significantly differ in the ocean component, despite sharing the same atmospheric (with different vertical resolution) and land components.

### Simulation setup

Our experimental setup includes a Reference (REF), and a multifactorial set of scenarios where AR and OAE are scaled up and/or combined, following the Shared Socioeconomic Pathway SSP3-7.0 emissions scenario. For every scenario, an ensemble of three realizations is run from 2015 to 2099 with each model. Both models are run in an emission-driven coupled ocean-land-atmosphere setup. In the REF simulation, land use, land management and related land-cover change (hereafter called "land use") remain constant at 2015 levels, and no land use transitions occur. Following Moustakis et al.[7], the dynamic vegetation module of JSBACH is switched off for both models, and thus there are no biogeographic changes in the cover fractions of the natural plant functional types (as could occur, e.g., in response to global warming), allowing for the full isolation of the AR effects on the Earth system.

In the AR simulation, land use follows the scenario developed by Moustakis et al.[7], which includes AR in the range of country pledges[61,62], reaching 595 Mha and 927 Mha of AR by 2060 and 2099, respectively. The employed AR pattern is based on information from a large number of IAM-generated scenarios[63], which are further constrained and disaggregated at the gridcell level guided by restoration potential[64,65] and ecosystem integrity[66] maps (see Moustakis et al.[7] for more details on scenario development).

To create an OAE scenario of comparable magnitude to AR, we apply as much alkalinity as is needed to roughly match the additional land sequestration in AR compared to REF. To do so, we first diagnose the dependency of the rate of $C_{ocean}$ uptake/Pmol of OAE on ambient $CO_2$ concentrations[15], by running a simulation similar to REF, where 0.17 Pmol/year of alkalinity are applied globally from 2015 onwards. In turn, the intermodel average ocean carbon uptake per Pmol alkalinity is estimated for every 10-year period, and is used to determine the average rate of alkalinity addition needed to match the cumulative additional AR sequestration on land for that period. This average rate is continuously applied during this period in the OAE scenario. In total, this adds up to 16 Pmol of alkalinity by 2099. Following the approach of Feng et al.[25], alkalinity is continuously and homogeneously applied over the ice-free coastline gridcells globally, which are considered to roughly correspond to the national Exclusive Economic Zones (EEZ) extending seaward 200 nautical miles (~370 km) from the coastline. OAE application close to the coastlines ensures proximity not only to the sea, but also to low-cost renewable electricity, and alkaline feedstock[60]. As evident in Fig. 2, ocean carbon sequestration under the OAE scenario successfully follows land carbon sequestration under AR.

To investigate the effects of scaling up AR, an additional scenario (halfAR) has been created, where half of AR is applied. In this scenario, the spatiotemporal characteristics of the applied AR pattern are not altered compared to the AR scenario, but rather every land use transition is merely halved in size. As a result, in the halfAR scenario 297.5 Mha and 463.5 Mha of forestation are reached by 2060 and 2099 respectively. Similarly for OAE, an additional scenario (halfOAE) has been created, where half of the alkalinity is applied at every timestep over the same gridcells, reaching 8 Pmol by 2099. Finally, to

investigate the joint effect of combining AR and OAE, a scenario (Mixed) has been created where both AR and OAE are employed. Similarly, a scenario (halfMixed) where both halfAR and halfOAE are employed has also been developed. A list of all scenarios and their characteristics is presented in Table 1.

With this setup, given that the REF and the various CDR scenarios only differ with respect to the application of CDR itself, the difference between any of the CDR scenarios and REF reflects the isolated effect of that particular CDR application on the Earth system. At the same time, the difference between halfAR and AR, halfOAE and OAE, and halfMixed and Mixed scenarios reflects the effect of scaling up CDR methods or portfolios. It should thus be noted that all carbon fluxes reported here are estimated as the difference of any given CDR scenario with respect to the mean REF carbon content. In particular, for every CDR scenario, we calculate the intermodel average difference in atmospheric ($C_{atmo}$), land ($C_{land}$), and ocean ($C_{ocean}$) carbon content compared to the REF scenario across time. Model-specific results are presented in the figures of the Supplementary Material.

### Linearity of carbon flux responses and removal efficiency

To assess the linearity of carbon fluxes in the scenarios where CDR is scaled up and/or combined, the deviation of the realized fluxes is compared to the fluxes obtained based on linearity expectations, as shown in detail in Table 1.

Removal efficiency (%) is expressed as the decrease in $C_{atmo}$ in a given CDR scenario compared to REF, divided by the increase in $C_{land}$ ($C_{ocean}$) under AR (OAE) compared to REF (Table 1). Removal efficiency is conceptually similar to the perturbation airborne fraction introduced by Jones et al.[4], however, as discussed in more detail in the main text, the denominator is the total change in $C_{land}$ ($C_{ocean}$) under AR (OAE), and not the carbon sequestration through AR (OAE) alone that should be excluding concurrent compensating fluxes within the land (ocean). Given this definition, removal efficiency is not estimated for the case of the Mixed and halfMixed scenarios.

Even though estimates of removal efficiency based on past studies are reported here, it should be noted that these have not been directly reported in these studies. These estimates are rather inferred from the data available in all publications. A table with the details on the estimates of removal efficiency based on previous studies is shown in the Supplementary Material (Table S1).

### Statistical treatment

For each model and scenario the statistical significance of changes in 2 m temperature during 2090–2099 (Fig. 5) at each gridcell is initially inferred at the 10% significance level. To do so, yearly mean temperature values from all 3 ensemble members during that period are pooled together, and a two-tailed Student's $t$ test adjusted to account for lag-1 temporal autocorrelation[67] is applied between the pooled data of the CDR and the REF scenario. However, when both models are considered, statistical significance is declared over gridcells where: (a) both models agree on the sign of the change, and (b) at least one model shows a statistically significant difference.

## Data availability
A repository with data supporting this publication has been published in Zenodo at: https://doi.org/10.5281/zenodo.15130372.

## Code availability
The Max Planck Institute's Earth System Model (MPI-ESM-1-2.01p7-LR) is made available under a version of the MPI-M software license agreement (the license and information on how to access the code can be found here: https://code.mpimet.mpg.de/projects/mpi-esm-license). FOCI is available at: https://zenodo.org/records/6772175. Python 3.11.2 has been used for all data analysis.

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

## Acknowledgements
This work was funded by the German Federal Ministry of Education and Research (BMBF), projects "CDRSynTra" (No. 01LS2101A) (Y.M., H.W., A.O., J.P.) and "STEPSEC" (01LS2102A) (T.N., J.P.). This work used resources of the Deutsches Klimarechenzentrum (DKRZ) granted by its Scientific Steering Committee (WLA) under project ID bm1241 for performing the MPI-ESM experiments and the analysis of the results. For performing the FOCI experiments, resources of the NHR Center Zuse Institute Berlin (ZIB) were used.

## Author contributions
Y.M.: Simulations MPI-ESM, Conceptualization, Methodology, Software, Validation, Formal analysis, Investigation, Data Curation, Writing—Original Draft, Review & Editing, Visualization. H.W.: Simulations FOCI, Conceptualization, Methodology, Data Curation, Writing—Review & Editing. T.N.: Methodology, Software, Writing—Review & Editing. A.O.: Conceptualization,Writing—Review & Editing, Project administration, Funding acquisition. J.P.: Conceptualization, Resources, Writing—Review & Editing, Supervision, Project administration, Funding acquisition.

## Funding

## Competing interests
The authors declare no competing interests.
