## [Peer Review file · Nature Communications]

No compromise in efficiency from the co-application of a marine and a terrestrial CDR method

Corresponding Author: Dr Yiannis Moustakis

Version 0:

Reviewer comments:

Reviewer #1

(Remarks to the Author)

Review of the manuscript "No compromise in efficiency from the co-application of marine and terrestrial 1 CDR" by Y. Moustakis et al.

The manuscript explores the efficiency of CDR for separate use and combination of two CR methods, namely afforestation/reforestation (AR) and ocean alkalinity enhancement (OAE). The manuscript is written clearly, concisely and logically. It is timely and important for broad audience. I find it suitable for publication in Nature Communications, pending minor revisions that address the suggested areas for improvement and clarification. Note, I will be happy to receive disagreeing responses to my comments (my understanding of methods and definitions is not complete), as long as your responses are persuading. Here, I encourage the authors to be extremely careful with definitions (e.g., CDR efficiency, removal efficiency etc.) and methods explanation.

Minor comments:

- 1) I am worried that the experiments in this study were designed in order to come to the conclusion on the linearity of scaling up the CDR technologies. When the half- or full- AR or OAE are applied, the difference in CDR application occurs on the subgrid level, and the scalability is expected. Furthermore, for AR, the 2099 year difference between half- and full-scale AR occurs under scenario with CO₂ concentration at near CO₂ fertilisation effect saturation levels that also work towards diminishing possible nonlinearity effects.
- 2) Theoretically, the considered CDR of the considered CDR methods can be defined as a function of carbon dioxide concentration and climatic factors, like surface temperature, wind speed etc. (e.g., such as in Gregory et al., 2009's carbon-concentration beta and carbon-climate gamma feedback framework). In the case of scaled or combined methods, because both CO₂ concentration and temperature are expected to be lower. Because at current levels of CO₂ and global warming the changes in carbon uptake are dominated by beta effects (i.e., changing CO₂ concentrations), the scaling/combining of CDR methods is expected to lead to saturation of the CDR. As I understand, this expectation is not proven by ESMS simulation results. Similarly, in their discussion, the authors state that "removal efficiency like depends on the emission trajectory" (line 130). However, scaling / combining CDR methods means increasing the atmospheric carbon removal, which affects emission trajectories. Then, the following statement on the insensitivity of removal efficiency to the rate of CDR application sounds contradicting. I would appreciate a more detailed discussion on this.
- 3) The atmospheric carbon removal by 2099 in Mixed scenario is 856 GtCO₂, while the sum of AR (429) and OAE (455) is 884 GtCO₂. This suggests somewhat a 3% difference in the linear expectations of additivity. Some discussion on the difference in these results would be appreciated.
- 4) While the manuscript has a very useful "temperature mitigation" section, it is rather descriptive in the current state. I suggest the authors to add more implications / discussion on linearity of scaling / combining CDR methods on temperature mitigation (as temperature mitigation is the main purpose of CDR application at the first place).
- 5) It would be beneficial for the paper if the last paragraph of Introduction was expanded by one-two sentences about the used methods/experiments (especially, if you refer to the Table 1 in Introduction).
- 6) Methods section "Models employed" would benefit from the description of the major differences and similarities of models. Do I understand correctly that only ocean module is different in two ESMS? Do they have identical land components?

Editorial comments:

- 1) Line 104 c is missing in "Cocean"
- 2) Line 242 "-" is missing before 0.14?

Reviewer #2

(Remarks to the Author)

The authors conducted dozens of simulations using two Earth system models to investigate how the combination and scaling-up of different CDR approaches may affect their efficiency of removing atmospheric CO₂. They showed that the efficiency of CDR was insensitive to the scale of CDR or the combinations of CDR. The manuscript is well written and will be a good contribution to the field.

My major concern is that the title of the manuscript overgeneralizes to land and ocean CDR, while only OAE and AR were tested here. Other CDR approaches, such as ocean iron fertilization, microalgae cultivation, ocean upwelling, etc., are also part of the CDR portfolio, as discussed in Keller et al. (2014). It remains unclear how these different approaches may interact, offset or complement each other.

As discussed in Section 7, OAE was homogeneously released over the whole coastal ocean, but the interannual variability in the ocean varies significantly among ocean regions, as reported by Yankovsky et al. (2024), a recent preprint in Biogeosciences. Therefore, the linearity of OAE will depend on the OAE locations.

How does the model resolution affect the linearity of scaling and combinations of CDR approaches? I think it warrants more discussion in Section 7.

Are there potential interactions or feedback mechanisms between OAE and AR across the land-ocean continuum?

Throughout the manuscript, I recommend using subscripts for "land" and "ocean" in "Cland" and "Cocean", respectively.

Figure 2. What are the reasons that the ocean shows much less variation compared to the land?

Line 104. "Cocean" is a typo.

Version 1:

Reviewer comments:

Reviewer #1

(Remarks to the Author)

The authors have addressed my comments satisfactorily, and the manuscript has been substantially improved. In my opinion, it is suitable for publication in its current form.

Reviewer #2

(Remarks to the Author)

The authors have addressed all my comments. I think the manuscript is in good shape for publication.

Response to reviewers

What follows is our detailed point-by-point responses to the reviewers' comments.

Reviewer #1 comments:

Comment:

The manuscript explores the efficiency of CDR for separate use and combination of two CR methods, namely afforestation/reforestation (AR) and ocean alkalinity enhancement (OAE). The manuscript is written clearly, concisely and logically. It is timely and important for broad audience. I find it suitable for publication in Nature Communications, pending minor revisions that address the suggested areas for improvement and clarification. Note, I will be happy to receive disagreeing responses to my comments (my understanding of methods and definitions is not complete), as long as your responses are persuading.

Response:

We thank the reviewer for their overall positive evaluation of our work.

Comment:

Here, I encourage the authors to be extremely careful with definitions (e.g., CDR efficiency, removal efficiency etc.) and methods explanation.

Response:

Based on the reviewer's suggestion we have now made appropriate changes throughout the text to ensure consistency. In the text we now only refer to "removal efficiency" (%), or "efficiency in mitigating warming" ($^{\circ}\text{C}/100 \text{ GtCO}_2$) to avoid any confusions. Since these metrics are introduced in the main text, the more general and intuitive term "efficiency of CDR" is used in the abstract.

We understand that what might have caused confusion was our reference in the Methods Section 10.3 stating that "several measures of the efficiency of CDR application are employed". This statement referred to the quantities C_{ocean} uptake/Pmol and C_{land} uptake/100 Mha, which, in an older version of this manuscript (pre-submission), were defined as "OAE-specific efficiency" and "AR-specific efficiency" respectively. This statement has been removed now.

Comment:

Minor comments:

1) I am worried that the experiments in this study were designed in order to come to the conclusion on the linearity of scaling up the CDR technologies. When the half- or full- AR or OAE are applied, the difference in CDR application occurs on the subgrid level, and the scalability is expected.

Response:

Scaling up the application of CDR indeed occurs at the subgrid level. This means that from halfAR to AR all additional forest is planted within the same gridcells, while the same also applies to additional alkalinity for OAE.

As indicated by the reviewer, linearity might not hold when scaling up the CDR deployment includes applying the additional alkalinity and/or establishing the additional forests over different regions. This is due to the fact that different ocean regions can yield different rates of C_{ocean} uptake/Pmol (Zhou et al., 2024) and, similarly, forests can have different capacities to sequester carbon across different regions (Egerer et al., 2024; Moustakis et al., 2024). In this case, inference on the linearity of the responses would be quite challenging, since our results would be dependent on the specific regions receiving the additional CDR. This means that, if regions more "efficient" in sequestering carbon were to be chosen, then likely a stronger-than-linear response would occur, while the choice of less "efficient" regions would similarly likely result in less-than-linear responses. It should also be noted that, apart from carbon sequestration, the application of AR can have different biogeophysical effects (e.g., on temperature) across different regions, which

commonly exhibit a strong latitudinal signal (warming in higher latitudes / cooling over the tropics) (De Hertog et al., 2023). Therefore, scaling up AR by applying it to different regions could bias the overall effect on temperature.

To avoid inducing such biases, when designing the study, we decided to define scaling up CDR solely as an increase in the rate of CDR application, while leaving all of its other characteristics unaltered. We also discuss these concepts in our responses to a comment made by Reviewer #2, based on the recent preprint of Yankovsky et al. (2024). This is now discussed in Lines [252-257].

Having clarified the above, it should be noted that scaling up CDR would not necessarily yield linear carbon cycle responses at the gridcell level, due to the multiple mechanisms that are at play. For example, as the forest cover increases at the gridcell level, plant water use and the evaporation of water intercepted by plant canopies increase, which suggest changes in water availability, affecting both the new and the existing forest. Most importantly, in our setup the land is not only affected by the atmosphere, but can also feed back to it. Therefore, changes in water fluxes, surface roughness and albedo can affect the partitioning of incoming net radiation to its sensible and latent heat components, and affect rainfall and temperature. Since such complex land-atmosphere interactions are at play, carbon sequestration on land at the gridcell level can deviate from linearity (Keller et al., 2018). This is in fact evident in Fig. 4. At the same time, emerging compensating fluxes emerge over gridcells where CDR is not applied over both the land and the ocean (e.g., the Amazon and the Southern Ocean) due to the reduction of atmospheric CO₂ concentrations, and it is unclear whether these should also scale linearly. Therefore, it remained unclear how all these feedbacks aggregate at the global level and whether they could result in non-linear carbon flux responses, especially when the effect of the reduction of atmospheric CO₂ concentrations due to the application of CDR itself is also considered. We tried to fill this gap with this work by employing fully coupled ESMs that represent all these feedbacks. Overall, the resulting linearity in terms of global C_{land} (C_{ocean}) uptake for AR (OAE) suggests that such feedbacks have indeed a rather weak effect when aggregating at the global level, which, however, is not merely the result of gridcell-level scalability.

It is important to highlight that our aim is not only to assess linearity in terms of C_{land} (C_{ocean}) uptake for AR (OAE), but also in terms of total C_{atmo} reduction, which corresponds to a constant removal efficiency (%). Our results demonstrated a constant removal efficiency that is insensitive to the magnitude of the CDR perturbation and there is no apparent reason suggesting that this should not hold when scaling up CDR includes different regions of deployment. This is now discussed in Lines [257-259].

Comment:

Furthermore, for AR, the 2099 year difference between half- and full-scale AR occurs under scenario with CO₂ concentration at near CO₂ fertilisation effect saturation levels that also work towards diminishing possible nonlinearity effects.

Response:

The reviewer is correct. In the high-emissions scenario employed here, the land and ocean carbon sink are less sensitive to changes in atmospheric CO₂, which likely works towards diminishing possible nonlinearity effects. We discussed this in the initial version of the manuscript (Lines [92-99] in new version), by examining the sensitivity of photosynthetic uptake and of C_{ocean} uptake/Pmol to changes in atmospheric CO₂ concentrations. This likely suggests that in scenarios with lower emissions and, thus, atmospheric CO₂ levels, the effect of CO₂ reduction induced by CDR itself can be relatively stronger, and therefore the linearity of responses could be distorted. Despite having already called for studying more emission trajectories in the Outlook section of the initial version of the manuscript, we now make this point more explicit in Lines [117-119].

Comment:

2) Theoretically, the considered CDR of the considered CDR methods can be defined as a function of carbon dioxide concentration and climatic factors, like surface temperature, wind speed etc. (e.g., such as in Gregory et al., 2009's carbon-concentration beta and carbon-climate gamma feedback framework). In the case of scaled or combined methods, because both CO₂ concentration and temperature are expected to be lower. Because at current levels of CO₂ and global warming the changes in carbon uptake are dominated by beta effects (i.e., changing CO₂ concentrations), the scaling/combining of CDR methods is expected to lead to saturation of the CDR. As I understand, this expectation is not proven by ESMs simulation results. Similarly, in their discussion, the authors state that "removal efficiency like

depends on the emission trajectory” (line 130). However, scaling / combining CDR methods means increasing the atmospheric carbon removal, which affects emission trajectories. Then, the following statement on the insensitivity of removal efficiency to the rate of CDR application sounds contradicting. I would appreciate a more detailed discussion on this.

Response:

The reviewer’s concern likely stems partly from how terminology was used in our study, so please let us clarify this and put it into context with past literature. According to Canadell et al. (2021) “*removal effectiveness (=removal efficiency as per our definition) of CDR is found to be slightly dependent on the rate and amount of CDR, and to be strongly dependent on the emissions scenario from which CDR is applied*”. According to Jones et al. (2016) “*the Perturbation Airborne Fraction (=removal efficiency as per our definition) from a given scenario is rather insensitive to the magnitude and timing of additional emissions*”. Clearly, in our paper the “*emission trajectory*” is what in these studies is referred to as “*(emissions) scenario*”. The “*trajectory*” or “*scenario*” refers to the emissions that force the models that can include: (a) the gross fossil fuel emissions and (b) negative emissions from technologies other than the ones imposed by us (if any). We initially avoided using the term “*scenario*” to avoid confusion with the multiple “*scenarios*” employed in our study (halfAR, AR, OAE etc.). However, to align with past literature and avoid confusion, we now also use the term “*scenario*”. In our understanding, the importance of the scenario/trajectory should mostly be understood through its role as the underlying “*background*” that determines the time-varying state and sensitivity of the Earth system to various perturbations, and not be interpreted as the resulting trajectory of net emissions including CDR.

However, the reviewer is right to point out that the application of CDR itself could potentially be of such a magnitude compared to the emissions scenario, thus having a significant effect. Due to the decrease in ambient atmospheric CO₂ concentrations, this would indeed likely emerge as a saturating response to the scaling up and/or combination of CDR, and one in theory could expect (a) a saturating C_{land},C_{ocean} uptake and (b) a decreasing removal efficiency. Regarding (a), we refer the reviewer to our responses above. Regarding (b), we had already discussed the case of Feng et al. (2017), where very strong CDR perturbations have indeed affected the removal efficiency, which has been shown to decrease with an increasing perturbation strength (Lines [143-148] in the new version). Nevertheless, uncertainty remains regarding the magnitude of CDR perturbation that can be considered strong enough to affect the removal efficiency. For example, Jones et al. (2016) reported small variations up to 5% when quadrupling the magnitude of CDR, despite it being even bigger than the cumulative emissions in the underlying RCP2.6 scenario. This is now discussed in Lines [148-150].

Comment:

3) The atmospheric carbon removal by 2099 in Mixed scenario is 856 GtCO₂, while the sum of AR (429) and OAE (455) is 884 GtCO₂. This suggests somewhat a 3% difference in the linear expectations of additivity. Some discussion on the difference in these results would be appreciated.

Response:

As the reviewer rightly points out, C_{atmo} reduction in the halfMixed and Mixed scenarios is ~0.4% and ~3% less than linear expectations respectively. We have now added to the manuscript that this is dominated by the variability of the terrestrial carbon sink (Fig. S5-7). However, we assume that what the reviewer implies is that this could point towards a tendency of the Earth system to yield potentially saturating responses in C_{atmo} reduction, in case strong enough perturbations are imposed to the Earth system that can starkly reduce ambient atmospheric CO₂ levels. We agree with the reviewer in their comment and this point is now also made in Lines [189-193].

Comment:

4) While the manuscript has a very useful “temperature mitigation” section, it is rather descriptive in the current state. I suggest the authors to add more implications / discussion on linearity of scaling / combining CDR methods on temperature mitigation (as temperature mitigation is the main purpose of CDR application at the first place).

Response:

We initially refrained from making more general comments on warming mitigation – especially at the gridcell level- due to the absence of large ensemble sizes, which would allow for robust statistical inference. Therefore, we initially maintained a rather more descriptive tone. However, we agree with the reviewer that our results warrant more discussion and have therefore expanded this section.

Overall, our results suggest that scaling up and/or combining CDR methods yields a roughly linear increase in the mitigation of global warming, which is the result of the nearly constant transient climate response to cumulative CO₂ emissions in scenarios with net positive emissions (MacDougall & Friedlingstein, 2015). However, this cannot hold at the gridcell-level, where the complex non-linear dynamics of both the biogeophysical and biogeochemical effects of CDR application on surface energy and moisture fluxes are at play (Amali et al., 2024; De Hertog et al., 2023, 2024). However, examining model-specific results (Fig. S10-11) suggests that patterns of biogeophysically-induced warming tendencies -even though statistically insignificant- tend to emerge also after scaling up AR or combining it with OAE, and are expressed either as a net warming or a weaker cooling (e.g., warming tendency over the Sahel region in MPI-ESM (Fig. S10)). This suggests that, despite temperature responses being non-linear, such features can still persist when scaling up and/or combining CDR methods. Nevertheless, in the absence of large ensemble sizes, temperature responses at the gridcell level can be masked by internal variability. This is now discussed in Lines [295-305].

Comment:

5) It would be beneficial for the paper if the last paragraph of Introduction was expanded by one-two sentences about the used methods/experiments (especially, if you refer to the Table 1 in Introduction).

Response:

Following the reviewer's suggestion, we have expanded the last paragraph of the Introduction (Lines [62-72]).

Comment:

6) Methods section "Models employed" would benefit from the description of the major differences and similarities of models. Do I understand correctly that only ocean module is different in two ESMs? Do they have identical land components?

Response:

The reviewer's understanding is correct. FOCI and MPI-ESM share an identical land component (JSBACH3) and the same atmospheric component (ECHAM6), albeit FOCI has more vertical atmospheric layers. The two models differ with regards to the ocean, both in terms of the models used, and in terms of the horizontal and vertical resolution of the ocean. This is now more clearly explained in the manuscript, where FOCI and MPI-ESM are separately presented first, and their differences are, in turn, highlighted more clearly (Lines [323-342]).

Comment:

Editorial comments:

1) Line 104 c is missing in "Cocean"

Response:

This has been corrected.

Comment:

2) Line 242 "- " is missing before 0.14?

Response:

Since we refer to "warming mitigation", or "cooling", or "temperature reduction", which already suggest a reduction in temperature, we now use positive numbers across the text instead.

Reviewer #2 comments:

Comment:

The authors conducted dozens of simulations using two Earth system models to investigate how the combination and scaling-up of different CDR approaches may affect their efficiency of removing atmospheric CO₂. They showed that the efficiency of CDR was insensitive to the scale of CDR or the combinations of CDR. The manuscript is well written and will be a good contribution to the field.

Response:

We thank the reviewer for their overall positive evaluation of our work.

Comment:

My major concern is that the title of the manuscript overgeneralizes to land and ocean CDR, while only OAE and AR were tested here. Other CDR approaches, such as ocean iron fertilization, microalgae cultivation, ocean upwelling, etc., are also part of the CDR portfolio, as discussed in Keller et al. (2014). It remains unclear how these different approaches may interact, offset or complement each other.

Response:

We understand the reviewer's concern. To avoid overgeneralization, we have rephrased the title to "No compromise in efficiency from the co-application of a marine and a terrestrial CDR method" which makes it clear that only one marine and one terrestrial CDR method have been used. Additionally, we now call for future studies to "further validate our results by exploring diverse portfolios that incorporate a broader range of CDR methods, assessing whether interactions between specific methods can occur, potentially offsetting linearity" (Lines [227-229]).

Comment:

As discussed in Section 7, OAE was homogeneously released over the whole coastal ocean, but the interannual variability in the ocean varies significantly among ocean regions, as reported by Yankovsky et al. (2024), a recent preprint in Biogeosciences. Therefore, the linearity of OAE will depend on the OAE locations.

Response:

We thank the reviewer for their comment and fully acknowledge that different regions have different OAE uptake efficiencies ($C_{\text{ocean}} \text{ uptake} / \text{Pmol}$) (Zhou et al., 2024), and that, consequently, different patterns of OAE application can yield different amounts of $C_{\text{ocean}} \text{ uptake} / \text{Pmol}$. Therefore, in cases where the scaling up of OAE application includes applying the additional alkalinity over different regions (and not over the same ocean gridcells as is the case here), then linearity could possibly be distorted. The same would likely hold for the case of AR, if e.g., we place new forest over different regions where they can sequester different amounts of carbon. We also discuss this in detail in our responses to the comments of Reviewer #1.

In their preprint, Yankovsky et al. (2024) suggest that Impulse Response Functions (IRFs) can be a powerful tool for making short-term predictions regarding the $C_{\text{ocean}} \text{ uptake}$ of OAE with a less than 7% error, which suggests a linear and time-invariant system. The authors suggest that a higher interannual variability violates the time invariance requirement of their approach, thus yielding higher errors. In fact, if an IRF that is agnostic of the effect of interannual variability on OAE uptake efficiency is considered, this can yield a ~10% error in the estimation of carbon uptake in the Labrador Sea, which shows a higher interannual variability (Lines 325-330 of Yankovsky et al.).

However, this is not entirely relevant to how we have approached scaling up the rate of OAE application. In our setup, the same regions were used for the half and the full application of CDR, to avoid confounding effects of regions' different characteristics, such as variability and efficiency. It should also be noted that to determine the amount of alkalinity needed for the C_{ocean} increase in the OAE scenario (compared to REF) to match the C_{land} increase in the AR scenario (compared to REF), we have run a diagnostic simulation where 0.17 Pmol/year of alkalinity are applied globally from 2015 onwards (Lines [358-365] in the new version). This allows us to calculate the time-varying globally averaged 10-year efficiency ($C_{\text{ocean}} \text{ uptake} / \text{Pmol applied}$) of OAE, which is mainly driven by the changes in ambient atmospheric CO₂ concentrations (Schwinger et al., 2024). Therefore, the possible effects of regional differences in interannual

variability and efficiency on the overall OAE uptake efficiency are already encapsulated in our initial average estimate. At the same time, as we discuss in the manuscript (Lines [60-61, 100-105] in the new version), there is no strong dependency of the efficiency ($C_{\text{ocean}} \text{ uptake} / \text{Pmol}$) on the rate of OAE application for the amounts of alkalinity considered here, and therefore linearity should not be jeopardized by scaling up OAE application over the same gridcells.

We comment on these in Lines [252-257].

Comment:

How does the model resolution affect the linearity of scaling and combinations of CDR approaches? I think it warrants more discussion in Section 7.

Response:

With respect to OAE: As we already discussed in the initial version of the manuscript (Lines [337-342] in the new version), our mini-ensemble including FOCI and MPI-ESM allows for inferring upon the influence of ocean resolution on linearity. This is due to the fact that the two models share the same atmospheric (albeit with a different number of vertical layers) and land component, but differ in terms of the ocean component and its resolution (please see our clarifications to Reviewer #1 above). Indeed, the rate of $C_{\text{ocean}} \text{ uptake} / \text{Pmol}$ applied can depend on various mechanisms affecting alkalinity diffusion, transport and mixing towards deeper sea layers and adjacent areas (He & Tyka, 2023) (Lines [339-341]). However, our results suggest that despite FOCI having a higher ocean resolution, it agrees with the coarser MPI-ESM with respect to the linearity of scaling up OAE and combining it with AR (Lines [247-248]).

With respect to AR: The differences between the MPI-ESM version used here (MPI-ESM1.2-LR) and one with a higher resolution (MPI-ESM1.2-HR) have been well documented in Müller et al. (2018) and can be insightful. Higher resolution improves, among others, the representation of North Atlantic storm tracks, blocking frequency and the North Atlantic Oscillation (Müller et al., 2018). We therefore fully acknowledge that employing high-resolution models would offer useful insight into the sensitivities and regional characteristics of carbon sequestration and feedback mechanisms, and therefore, linearity at the gridcell level. However, even though averaging climatic conditions within coarser gridcells tends to smooth out climatic extremes, thus resulting in a “greener” world (Müller & Lucht, 2007), the temporal dynamics of the terrestrial carbon sink remain qualitatively the same, regardless of the spatial resolution of the model. It should be noted that, when increasing model resolution, the process representation in the land component does not change, and therefore so does the sensitivity of carbon sequestration to ambient atmospheric CO_2 concentrations (which is the dominant factor driving the responses in our models) and climate. This is discussed now in Lines [259-263]. Nevertheless, it should be noted that we don’t claim linearity at the gridcell level (as also evident in Fig. 4), but rather only suggest that linear approximations could be used as a “*first order approximation*” (Lines [240-245]).

Comment:

Are there potential interactions or feedback mechanisms between OAE and AR across the land-ocean continuum?

Response:

In FOCI and MPI-ESM, the only direct interaction between the land and the ocean that is not mediated by the atmosphere is the influx of freshwater to the ocean. Changes in the freshwater flux could affect the salinity, stratification, and mixing of the the sea. Our results suggest that at the global level there is little interaction between AR and OAE, despite the biogeophysical effects of AR on hydroclimatic variability and potential changes in the freshwater flux into the ocean, which are rather lower-order effects. We comment on this in Lines [193-195]. The direct exchange between the land and the ocean of carbon taken up through photosynthesis and chemical weathering is not represented in the models employed here, as is the case in most Earth System Models (Regnier et al., 2013, 2022).

Comment:

Throughout the manuscript, I recommend using subscripts for “land” and “ocean” in “Cland” and “Cocean”, respectively.

Response:

We have made changes based on the reviewer's suggestion.

Comment:

Figure 2. What are the reasons that the ocean shows much less variation compared to the land?

Response:

The reviewer is correct in observing that the changes in oceanic carbon are less variable than the changes in land carbon. A significant portion of the variability of the ocean carbon sink has been historically determined by the growth rate of atmospheric pCO₂ (McKinley et al., 2020) and the ocean carbon sink is generally indeed known to show less variability than the terrestrial carbon sink, which is more strongly dependent on climatic variability (Crisp et al., 2022; Friedlingstein et al., 2024).

Comment:

Line 104. "Coean" is a typo.

Response:

This has been corrected.

References

- Amali, A. A., Schwingshackl, C., Ito, A., Barbu, A., Delire, C., Peano, D., Lawrence, D. M., Wårlind, D., Robertson, E., Davin, E. L., Shevliakova, E., Harman, I. N., Vuichard, N., Miller, P. A., Lawrence, P. J., Ziehn, T., Hajima, T., Brovkin, V., Zhang, Y., ... Pongratz, J. (2024). Biogeochemical versus biogeophysical temperature effects of historical land-use change in CMIP6. (*Accepted*) *EGU Earth System Dynamics*, 1–55. <https://doi.org/10.5194/egusphere-2024-2460>
- Canadell, J. G., Monteiro, P. M. S., Costa, M. H., Cotrim da Cunha, L., Cox, P. M., Eliseev, A. V., Henson, S., Ishii, M., Jaccard, S., Koven, C., Lohila, A., Patra, P. K., Piao, S., Rogelj, J., Syampungani, S., Zaehle, S., & Zickfeld, K. (2021). Global Carbon and other Biogeochemical Cycles and Feedbacks. In *Climate Change 2021: The Physical Science Basis. Contribution of Working Group I to the Sixth Assessment Report of the Intergovernmental Panel on Climate Change* [Masson-Delmotte, V., P. Zhai, A. Pirani, S.L. Connors, C. Péan, S. Berger, N. Caud, Y. Chen, L. Goldfarb, M.I. Gomis, M. Huang, K. Leitzell, E. Lonnoy, J.B.R. Matthews, T.K. Maycock, T. Waterfield, O. Yelekçi, R. Yu, and B. Zhou (eds.)] (pp. 673–816). Cambridge University Press. 10.1017/9781009157896.007
- Crisp, D., Dolman, H., Tanhua, T., McKinley, G. A., Hauck, J., Bastos, A., Sitch, S., Eggleston, S., & Aich, V. (2022). How Well Do We Understand the Land-Ocean-Atmosphere Carbon Cycle? *Reviews of Geophysics*, 60(2), e2021RG000736. <https://doi.org/10.1029/2021RG000736>
- De Hertog, S. J., Havermann, F., Vanderkelen, I., Guo, S., Luo, F., Manola, I., Coumou, D., Davin, E. L., Duveiller, G., Lejeune, Q., Pongratz, J., Schleussner, C.-F., Seneviratne, S. I., & Thiery, W. (2023). The biogeophysical effects of

idealized land cover and land management changes in Earth system models. *Earth System Dynamics*, 14(3), 629–667. <https://doi.org/10.5194/esd-14-629-2023>

De Hertog, S. J., Lopez-Fabara, C. E., van der Ent, R., Keune, J., Miralles, D. G., Portmann, R., Schemm, S., Havermann, F., Guo, S., Luo, F., Manola, I., Lejeune, Q., Pongratz, J., Schleussner, C.-F., Seneviratne, S. I., & Thiery, W. (2024). Effects of idealized land cover and land management changes on the atmospheric water cycle. *Earth System Dynamics*, 15(2), 265–291. <https://doi.org/10.5194/esd-15-265-2024>

Egerer, S., Falk, S., Mayer, D., Nützel, T., Obermeier, W. A., & Pongratz, J. (2024). How to measure the efficiency of bioenergy crops compared to forestation. *Biogeosciences*, 21(22), 5005–5025. <https://doi.org/10.5194/bg-21-5005-2024>

Feng, E. Y., Koeve, W., Keller, D. P., & Oschlies, A. (2017). Model-Based Assessment of the CO₂ Sequestration Potential of Coastal Ocean Alkalinization. *Earth's Future*, 5(12), 1252–1266. <https://doi.org/10.1002/2017EF000659>

Friedlingstein, P., O'Sullivan, M., Jones, M. W., Andrew, R. M., Hauck, J., Landschützer, P., Le Quéré, C., Li, H., Luijckx, I. T., Olsen, A., Peters, G. P., Peters, W., Pongratz, J., Schwingshackl, C., Sitch, S., Canadell, J. G., Ciais, P., Jackson, R. B., Alin, S. R., ... Zeng, J. (2024). Global Carbon Budget 2024. *Earth System Science Data Discussions*, 1–133. <https://doi.org/10.5194/essd-2024-519>

He, J., & Tyka, M. D. (2023). Limits and CO₂ equilibration of near-coast alkalinity enhancement. *Biogeosciences*, 20(1), 27–43. <https://doi.org/10.5194/bg-20-27-2023>

Jones, C. D., Ciais, P., Davis, S. J., Friedlingstein, P., Gasser, T., Peters, G. P., Rogelj, J., Vuuren, D. P. van, Canadell, J. G., Cowie, A., Jackson, R. B., Jonas, M., Kriegler, E., Littleton, E., Lowe, J. A., Milne, J., Shrestha, G., Smith, P., Torvanger, A., & Wiltshire, A. (2016). Simulating the Earth system response to negative emissions. *Environmental Research Letters*, 11(9), 095012. <https://doi.org/10.1088/1748-9326/11/9/095012>

Keller, D. P., Lenton, A., Littleton, E. W., Oschlies, A., Scott, V., & Vaughan, N. E. (2018). The Effects of Carbon Dioxide Removal on the Carbon Cycle. *Current Climate Change Reports*, 4(3), 250–265. <https://doi.org/10.1007/s40641-018-0104-3>

MacDougall, A. H., & Friedlingstein, P. (2015). The Origin and Limits of the Near Proportionality between Climate Warming and Cumulative CO₂ Emissions. *Journal of Climate*, 28(10), 4217–4230. <https://doi.org/10.1175/JCLI-D-14-00036.1>

- McKinley, G. A., Fay, A. R., Eddebbar, Y. A., Gloege, L., & Lovenduski, N. S. (2020). External Forcing Explains Recent Decadal Variability of the Ocean Carbon Sink. *AGU Advances*, 1(2), e2019AV000149. <https://doi.org/10.1029/2019AV000149>
- Moustakis, Y., Nützel, T., Wey, H.-W., Bao, W., & Pongratz, J. (2024). Temperature overshoot responses to ambitious forestation in an Earth System Model. *Nature Communications*, 15(1), 8235. <https://doi.org/10.1038/s41467-024-52508-x>
- Müller, C., & Lucht, W. (2007). Robustness of terrestrial carbon and water cycle simulations against variations in spatial resolution. *Journal of Geophysical Research: Atmospheres*, 112(D6). <https://doi.org/10.1029/2006JD007875>
- Müller, W. A., Jungclaus, J. H., Mauritsen, T., Baehr, J., Bittner, M., Budich, R., Bunzel, F., Esch, M., Ghosh, R., Haak, H., Ilyina, T., Kleine, T., Kornblueh, L., Li, H., Modali, K., Notz, D., Pohlmann, H., Roeckner, E., Stemmler, I., ... Marotzke, J. (2018). A Higher-resolution Version of the Max Planck Institute Earth System Model (MPI-ESM1.2-HR). *Journal of Advances in Modeling Earth Systems*, 10(7), 1383–1413. <https://doi.org/10.1029/2017MS001217>
- Regnier, P., Friedlingstein, P., Ciais, P., Mackenzie, F. T., Gruber, N., Janssens, I. A., Laruelle, G. G., Lauerwald, R., Luysaert, S., Andersson, A. J., Arndt, S., Arnosti, C., Borges, A. V., Dale, A. W., Gallego-Sala, A., Goddérís, Y., Goossens, N., Hartmann, J., Heinze, C., ... Thullner, M. (2013). Anthropogenic perturbation of the carbon fluxes from land to ocean. *Nature Geoscience*, 6(8), 597–607. <https://doi.org/10.1038/ngeo1830>
- Regnier, P., Resplandy, L., Najjar, R. G., & Ciais, P. (2022). The land-to-ocean loops of the global carbon cycle. *Nature*, 603(7901), 401–410. <https://doi.org/10.1038/s41586-021-04339-9>
- Schwinger, J., Bourgeois, T., & Rickels, W. (2024). On the emission-path dependency of the efficiency of ocean alkalinity enhancement. *Environmental Research Letters*. <https://doi.org/10.1088/1748-9326/ad5a27>
- Yankovsky, E., Zhou, M., Tyka, M., Bachman, S., Ho, D., Karspeck, A., & Long, M. (2024). Impulse response functions as a framework for quantifying ocean-based carbon dioxide removal. *EGUsphere*, 1–26. <https://doi.org/10.5194/egusphere-2024-2697>
- Zhou, M., Tyka, M. D., Ho, D. T., Yankovsky, E., Bachman, S., Nicholas, T., Karspeck, A. R., & Long, M. C. (2024). Mapping the global variation in the efficiency of ocean alkalinity enhancement for carbon dioxide removal. *Nature Climate Change*, 1–7. <https://doi.org/10.1038/s41558-024-02179-9>